# Artificial intelligence-assisted prostate cancer diagnosis for reduced use of immunohistochemistry
Anders Blilie [1,2,13], Nita Mulliqi[3,13], Xiaoyi Ji[3], Kelvin Szolnoky[3], Sol Erika Boman [3,4], Matteo Titus [3], Geraldine Martinez Gonzalez[3], José Asenjo[5], Marcello Gambacorta[6], Paolo Libretti[6], Einar Gudlaugsson[1], Svein R. Kjosavik [7,8], Lars Egevad[9], Emiel A. M. Janssen[1,10,11], Martin Eklund [3] & Kimmo Kartasalo [12] ✉

## Abstract

**Background:** Prostate cancer diagnosis heavily relies on histopathological evaluation, which is subject to variability. While immunohistochemical staining (IHC) assists in distinguishing benign from malignant tissue, it increases workload, costs, and leads to diagnostic delays. Artificial intelligence (AI) presents a promising solution to reduce reliance on IHC by accurately classifying atypical glands and borderline morphologies in hematoxylin and eosin (H&E) stained tissue sections. **Methods:** In this study, we evaluated an AI model's ability to minimize IHC use without compromising diagnostic accuracy. We retrospectively analyzed prostate core needle biopsies from routine diagnostics at three different pathology sites. These cohorts consisted exclusively of diagnostically challenging cases where pathologists had required IHC to finalize the diagnosis. **Results:** We show that the AI model achieves high performance, with area under the curve values ranging from 0.951 to 0.993 for detecting cancer in routine H&E-stained slides. When applying sensitivity-prioritized diagnostic thresholds, the model reduces the need for IHC staining by 44.4%, 42.0%, and 20.7% across the three cohorts, without a single false negative prediction. Among slides with a benign ground truth label, IHC use is reduced by up to 80.6%. **Conclusions:** This AI model shows promise for reducing unnecessary IHC use in difficult prostate biopsy cases while maintaining diagnostic safety. Its integration into clinical workflows could streamline decision-making in prostate pathology and alleviate resource burdens.

## Plain language summary

Diagnosing prostate cancer typically involves examining tissue samples under a microscope. In challenging cases, doctors often use a special test called immunohistochemistry (IHC) to help confirm whether cancer is present. However, IHC adds time, cost, and extra work to the diagnostic process. In this study, we tested an artificial intelligence (AI) tool to see if it could accurately identify prostate cancer using only standard tissue images—without needing IHC. We analyzed especially difficult biopsy cases from three different hospitals, where pathologists had originally needed IHC to make a diagnosis. The AI tool was highly accurate and, when using a safety-first approach, it could reduce the use of IHC by 20% to 44% depending on the site without missing any cancers. Importantly, when focusing only on slides that were ultimately benign, the AI could reduce IHC use by up to 80.6%. This suggests AI could help pathologists make faster and more efficient decisions while maintaining diagnostic safety.

Histopathological evaluation of prostate biopsies using the Gleason grading system is a cornerstone in the diagnosis and management of prostate cancer[1–3]. However, Gleason grading is notoriously subjective, showing high inter- and intraobserver variability, resulting in over- and underdiagnosis[4–7]. To standardize diagnostics, the International Society of Urological Pathology (ISUP) updated grading guidelines for prostate cancer to convert Gleason scores into ISUP grades (also called 'grade groups') from 1 to 5[8]. Pathological assessment can be aided by immunohistochemical staining (IHC), which in prostate pathology is mainly used for the identification of prostatic basal cells. These cells are present around the periphery of benign glandular structures but are lost in the development of prostatic adenocarcinoma (with rare exceptions), making absent basal-cell IHC staining strongly suggestive of malignancy[9–12]. ISUP recommends using basal-cell IHC markers to confirm cancer when encountering small foci of atypical glands where a definitive malignant diagnosis cannot be rendered based on hematoxylin & eosin (H&E) staining[9]. The IHC markers (antibodies) most commonly used to identify prostatic basal cells are high-molecular-weight cytokeratin (HMWCK) and p63, often used together to increase sensitivity[9].

---

In rare cases, non-cancerous morphological variants such as adenosis, atrophy, or intraepithelial neoplasia can have areas of absent basal cell staining, or conversely, prostate cancer can paradoxically exhibit positive IHC staining for basal cell markers[9]. Thus, IHC expression must be interpreted carefully and correlated with H&E morphology, which can be challenging.

The decision to order IHC for a given tissue block is inherently subjective, depending on the judgment of the pathologist. Differences in uropathology experience, combined with high observer variability, naturally lead to varying practices for ordering IHC[13]. Personal preferences also play a role, as some pathologists rely on IHC as a safety net to minimize misdiagnosing malignancy even when morphological suspicion is low. This variation extends across pathology laboratories, where many IHC investigations ultimately result in a benign diagnosis[14]. Furthermore, some laboratories are known to preemptively order IHC for all prostate biopsies, anticipating a high likelihood of IHC requests from pathologists. The use of IHC incurs costs in both time and resources. Each antibody reagent has a per-use price, and every tissue block requiring IHC must be re-cut, stained, and further processed. This places additional strain on the pathology lab and extends turnaround times, ultimately delaying the final diagnosis[15,16].

Transitioning from glass slides to digital whole-slide images (WSI) is widely considered the third revolution in modern pathology, following the introduction of IHC and the inception of genomic medicine using molecular-based methods[17]. Artificial intelligence (AI) has shown potential in standardizing histopathological grading of prostate cancer[6,18–20], as well as in predicting treatment response[21] and patient outcomes[22]. Recently, pathology foundation models (FM) have shown promise in pan-cancer detection[23–26]. Despite the growing role of AI in pathology and its potential to enhance diagnostic consistency, there remains a critical gap in leveraging AI models to systematically standardize and minimize unnecessary IHC usage in routine prostate cancer diagnostics. A previous study proposed an AI solution for identifying tissue blocks that are likely to require IHC and preemptively order IHC prior to pathologist evaluation[16]. Another study found that retrospective evaluation of prostate biopsies using an AI model led to a reduced need for IHC compared to the traditional diagnostic approach[27]. IHC staining has also been used as the reference standard in a study developing an AI model for prostate tissue segmentation[28]. While these studies have aimed to replicate pathologists' IHC-ordering patterns for workflow optimization, assess IHC frequency in a research setting, or improve tissue segmentation, to our knowledge, no study has used AI models to standardize IHC usage in prostate pathology or minimize IHC requests for benign slides in routine clinical practice.

We utilize an AI model trained on prostate core needle biopsies for prostate cancer diagnosis and Gleason grading, which demonstrates robust performance in handling challenging tissue morphologies[29]. We hypothesize that such a model, capable of accurately diagnosing small foci with atypical glands and borderline morphology, can reduce reliance on IHC in routine practice. This study follows a pre-specified protocol, detailing study design and patient cohorts[30]. In this study, we apply the model to WSIs of H&E-stained prostate biopsies from multiple international pathology sites. These WSIs represent slides where the diagnosing pathologist required IHC, in addition to H&E staining, to render a final diagnosis. We apply sensitivity-prioritized diagnostic thresholds to minimize false negative predictions—critical for ensuring no cancers are overlooked—while maintaining the high specificity required to effectively reduce IHC usage for benign slides. We show that for true negative slides, where the pathologist's suspicion of malignancy is low, AI-based support can potentially eliminate a substantial number of IHC investigations traditionally used for rule-out purposes.

## Methods
### Study design
The full dataset underlying the AI models of this study comprises biopsy samples from 7243 patients across 15 clinical sites in 11 countries, encompassing 58,744 physical glass slides containing ~100,000 biopsy cores.

Slides were digitized using 14 scanners (nine different models from five manufacturers), producing a total of 82,584 WSIs. For this study, we exclusively included slides where IHC staining targeting basal cells was performed in the diagnostic process, considering this to be a surrogate marker for the pathologist not being able to establish infiltration status by H&E-staining alone. To avoid data leakage and ensure robust generalization, only held-out test data from patients who were not part of AI model training or hyperparameter tuning were included in the final analysis. The patient sampling strategy for internal and external validation was pre-specified in the study protocol, ensuring a structured and reproducible selection process[30]. Among the 15 patient cohorts represented in the dataset, only three cohorts had reliable information regarding IHC staining status: Stavanger University Hospital (SUH), Synlab France (SFR), and Synlab Switzerland (SCH). The clinical characteristics of the included patients are summarized in Table 1 and the CONSORT diagram outlining the sample inclusion process is provided in Supplementary Fig. 1.

### Data cohorts
**Cohort 1: Stavanger University Hospital**. The SUH samples represent consecutive cases collected from routine diagnostics at the Department of Pathology, Stavanger University Hospital, Norway, between December 2016 and March 2018. Biopsies were obtained at the Department of Urology, Stavanger University Hospital, as well as private urological clinics within the same geographic region. Most biopsies were transrectal and systematic, with a minority involving MRI-guided targeted biopsy collection. The slides were digitized with a Hamamatsu S60 scanner (40×, pixel size 0.2199 μm).

Tabulated slide-level information from the SUH cohort contained IHC status, including the type of stain used, as well as Gleason scores and cancer length per slide. As portions of this cohort were used in the development of the AI model, only slides reserved for internal validation were considered for inclusion. From this subset, all slides where a basal-cell IHC marker was used, almost invariably HMWCK (CK903/34βE12), were included in the study (n = 234; 129 benign, 105 malignant). 12 different diagnosing pathologists were represented in this subset of samples. Detailed information on the IHC and H&E staining protocols, including antibody clones, equipment, and site-specific procedures, was not available.

**Cohort 2: Synlab France**. The SFR cohort comprises consecutive cases collected from routine diagnostics at the Technipath-Synlab Medical Laboratory in Dommartin, Rhône, France, between September 2020 and December 2020. This cohort was entirely external, i.e., not used in AI model development. The samples were a mixture of systematic transrectal biopsies and MRI-guided targeted biopsies. The slides were digitized with a Philips IntelliSite Ultra Fast Scanner (40×, pixel size 0.2500 μm, the same device as for the cohort SCH).

From the SFR cohort, we also had slide-level information regarding the use of IHC. Gleason scores and cancer length for individual slides were available; however, there was no tabulated specification of the type of IHC stain(s) performed. To determine this, we manually investigated de-identified pathology reports to extract the missing information. All slides where a basal-cell marker was used, almost invariably p63 in combination with P504S/AMACR, were included in our study (n = 112; 66 benign, 46 malignant). The pathologists' names were redacted in the reports, and thus, we could not determine the number of pathologists represented in this subset of samples. Detailed information on the IHC and H&E staining protocols, including antibody clones, equipment, and site-specific procedures, was not available.

**Cohort 3: Synlab Switzerland**. The SCH samples represent consecutive cases collected from routine diagnostics at the Argot Laboratory in Lausanne, Switzerland, between January 2020 and December 2020. This dataset was entirely external, i.e., not used in AI-model development. Biopsies were a mixture of systematic transrectal biopsies and MRI-guided targeted biopsies. The slides were digitized with a Philips

**Table 1 | Clinical characteristics of patient cohorts**

| Cohorts | SUH | SFR | SCH |
|---|---|---|---|
| Dataset type | Internal | External | External |
| Pathologists represented | 12 | N/A | 5 |
| Scanner model | Hamamatsu NanoZoomer S60 | Philips IntelliSite UFS | Philips IntelliSite UFS |
| **Number of patients** | **_n_ = 99** | **_n_ = 49** | **_n_ = 75** |
| Age (years) | | | |
| ≤49 years | 1 (1.0%) | 0 (0.0%) | 0 (0.0%) |
| 50–54 years | 3 (3.0%) | 2 (4.1%) | 1 (1.3%) |
| 55–59 years | 9 (9.1%) | 8 (16.3%) | 5 (6.7%) |
| 60–64 years | 25 (25.3%) | 4 (8.2%) | 10 (13.3%) |
| 65–69 years | 23 (23.2%) | 14 (28.6%) | 16 (21.3%) |
| ≥70 years | 38 (38.4%) | 21 (42.8%) | 43 (57.4%) |
| Prostate-specific antigen (ng/ml) | | | |
| 0–3 ng/mL | 7 (7.1%) | 1 (2.0%) | 1 (1.3%) |
| >3–5 ng/mL | 15 (15.1%) | 4 (8.2%) | 6 (8.0%) |
| >5–10 ng/mL | 50 (50.5%) | 29 (59.2%) | 18 (24.0%) |
| ≥10 ng/mL | 26 (26.3%) | 8 (16.3%) | 13 (17.3%) |
| «Elevated» | 0 (0.0%) | 0 (0.0%) | 8 (10.7%) |
| Missing | 1 (1.0%) | 7 (14.3%) | 29 (38.7%) |
| **Number of WSIs** | **_n_ = 234** | **_n_ = 112** | **_n_ = 164** |
| ISUP grades (Gleason scores) | | | |
| Benign | 129 (55.1%) | 66 (58.9%) | 65 (39.6%) |
| ISUP 1 (3 + 3) | 60 (25.6%) | 41 (36.6%) | 50* (30.5 %) |
| ISUP 2 (3 + 4) | 15 (6.4%) | 3 (2.7%) | 17* (10.4%) |
| ISUP 3 (4 + 3) | 10 (4.3%) | 1 (0.9%) | 24* (14.6%) |
| ISUP 4 (4 + 4, 3 + 5, 5 + 3) | 9 (3.9%) | 1 (0.9%) | 3* (1.8%) |
| ISUP 5 (4 + 5, 5 + 4, 5 + 5) | 11 (4.7%) | 0 (0.0%) | 5* (3.1%) |
| Cancer length (mm) | | | |
| No cancer | 129 (55.1%) | 66 (58.9%) | 65 (39.6%) |
| >0–1 mm | 23 (9.8%) | 1 (0.9%) | 3* (1.8%) |
| >1–5 mm | 42 (18.0%) | 21 (18.8%) | 30* (18.3%) |
| >5–10 mm | 15 (6.4%) | 6 (5.4%) | 8* (4.9%) |
| >10 mm | 25 (10.7%) | 13 (11.6%) | 57* (34.8%) |
| Missing | 0 (0.0%) | 5 (4.4%) | 1 (0.6%) |

An overview of the patients and slides included in the three cohorts, including age, PSA, cancer grade, and cancer length distributions. Scanners used for digitization and the number of pathologists (unavailable for SFR) represented within each cohort are specified. *In the SCH cohort, cancer grading and length were assigned at the location level (across multiple slides), meaning information for individual WSIs is unavailable. Instead, we report the overall grade and cancer length assigned to the location. *ISUP* International Society of Urological Pathology grade, *PSA* prostate-specific antigen, *SCH* Synlab Laboratory Switzerland, *SFR* Synlab Laboratory France, *SUH* Stavanger University Hospital, *WSI* whole-slide image.

IntelliSite Ultra Fast Scanner (40×, pixel size 0.2500 μm, the same device as for the cohort SFR).

The SCH cohort differs from the other cohorts in that the diagnoses were reported in a pooled manner for each anatomical location of the prostate sampled with multiple biopsy cores. The combined Gleason score per location covered by multiple slides was reported. Consequently, no tabulated slide-level information regarding Gleason score or cancer length was available for this cohort. The data tables contained location-level information regarding IHC use but did not specify which particular slide(s) had IHC requested or the type of stain used. Getting this information required reading the de-identified pathology reports. For cases involving IHC, the reports detailed the specific slides where IHC was requested and whether they were benign or malignant, enabling us to include only relevant slides in our study. To ensure validity, it was necessary to confirm a systematic order linking scanned slides to their corresponding location-level report information. Approximately 150 WSIs were evaluated by our study pathologist (A.B.), verifying that such a systematic order existed (e.g., that

"Slide 2 C" in a report corresponded to "Scan 3" from "Location 2"). The reports also specified the type of IHC used, which was almost invariably p63 (often in combination with P504S/AMACR). After this filtering process, all IHC slides were included in our study (_n_ = 164; 65 benign, 99 malignant). Five different diagnosing pathologists were represented in this subset of samples. Detailed information on the IHC and H&E staining protocols, including antibody clones, equipment, and site-specific procedures, was not available.

**Tissue detection and tiling**

Tissue detection from WSIs was performed using a custom-built tissue segmentation model based on a UNet++ architecture, incorporating a ResNeXt-101 (32 × 4d) encoder[31]. Initially, 512 × 512 px patches were extracted across the entire WSI at 8.0 μm/px resolution, with a 128 px overlap, followed by pixel-wise segmentation to identify tissue regions. These segmented regions were then combined into a single binary tissue mask per WSI. Next, 256 × 256 px high-resolution tissue patches were

extracted at 1.0 μm/px resolution, using the segmentation masks to retain only those patches where at least 10% of pixels contained tissue. During model training, patches were extracted without overlap to reduce GPU memory usage, whereas, for model prediction, a 128 px overlap was used to enhance diagnostic accuracy. To achieve a resolution of 1.0 μm/px, patches were downsampled from the nearest higher resolution level in the WSI resolution pyramid using Lanczos resampling. Extracted patches were stored in the TFRecord format for efficient disk storage, with each WSI saved as a separate file.

## AI model

The task-specific AI model used for evaluation in this study was trained on digitized prostate core needle biopsies for prostate cancer diagnosis and Gleason grading[29]. The model was built using an attention-based multiple instance learning (ABMIL) architecture with weakly supervised learning, leveraging only slide-level labels. The model uses an EfficientNet-V2-S encoder[32] to extract patch-level feature embeddings that are further aggregated into slide-level representations with the ABMIL and classification layers providing classification of the two Gleason patterns (i.e., 3, 4, or 5), further translated into Gleason score and ISUP grade. The grading model was trained in an end-to-end fashion where all model parameters were jointly optimized for cross-entropy loss using the AdamW optimizer[33] with a base learning rate of 0.0001. Further details regarding design choices, hyperparameters, and validation results can be found in the original publication[29]. UNI[25] and Virchow2[34] foundation models were used within the same training pipeline; however, the weights of the encoders were kept frozen, and only the ABMIL and subsequent classification layers were trained identically to the task-specific model. The model was trained on 10 cross-validation folds stratified by the patient and ISUP grade. During model predictions, test-time augmentation (TTA) was applied for three iterations per model, and the final prediction was obtained as the majority vote of the 30 predicted Gleason scores (10 models × 3 TTA runs), and further translated into an ISUP grade. Cancer probability was obtained as the median over the ensemble. Mean attention scores from the ABMIL models were used for each tile to highlight regions of interest that the AI focused on for the final diagnosis.

## Statistics and reproducibility

Using the numerical value for AI-predicted cancer probability of a given WSI, we analyze the results for each cohort at different model operating points (i.e., different thresholds for a positive prediction), allowing us to prioritize either sensitivity or specificity for cancer detection. We analyzed the data using thresholds ranging from 0.5 to 0.01. To quantify the concordance of negative/positive diagnoses for prostate cancer with the reference standard, we used sensitivity (true positive rate), specificity (true negative rate), and AUC. All reported values are point estimates. The statistical calculations were conducted using the Python modules NumPy (v1.24.0), scikit-learn (v1.2.2), and Pandas (v1.5.3). All computational analyses were verified to be deterministic and, as such, fully repeatable. The analyzed material consisted of routine clinical samples without biological replicates.

## Hardware and software

Model training and predictions were run as described earlier[29]. We used Python (v3.8.10), PyTorch (v2.0.0, CUDA 12.2) (https://pytorch.org), and PyTorch DDP for multi-GPU training for all experiments across all models. We used the pre-trained weights for UNI and Virchow2 FMs from their official releases on the HuggingFace hub (https://huggingface.co/MahmoodLab/UNI; https://huggingface.co/paige-ai/Virchow2) and integrated them with the ViT implementations provided by the timm library (v0.9.8). All experiments were done on two high-performance clusters: Alvis (part of the National Academic Infrastructure for Supercomputing in Sweden) and Berzelius (part of the National Supercomputer Centre). On Alvis, training was done on 4 × 80GB NVIDIA A100 GPUs (256 GB system memory, 16 CPU cores per GPU). On Berzelius, training was done on 8 × 80

GB NVIDIA A100 GPUs (128 GB system memory, 16 CPU cores per GPU). Predictions were run on the clusters on a single 40 GB A100 NVIDIA GPU. Docker (v20.10.21) was used locally, Singularity and Apptainer were used on the computing clusters. OpenSlide (v4.0.0), openslide-python (v1.3.1), and OpenPhi (v2.1.0) were used to access WSIs. DareBlopy (v0.0.5) was used for compatibility between the TFRecord data format (.tfrecord) and PyTorch. Albumentations (v1.3.1) and Stainlib (v0.6.1) were used for image augmentations. For implementing the tissue segmentation model PyTorch segmentation_models_pytorch library (v0.3.3) was used. NumPy (v1.24.0), scikit-learn (v1.2.2), and Pandas (v1.5.3) were used for numerical operations, model evaluation, and data management. Pillow (v9.4.0) and OpenCV-python were used for basic image processing tasks. Matplotlib (v3.7.1) and Seaborn (v0.12.2) were used for plots and figures, and Biorender was used to assemble figure panels. Pathologists' reviews of false negative cases were done using QuPath (v0.4.3)[35].

## Ethical considerations

This study included data gathered in one or more collection rounds at participating international sites between 2012 and 2024. All datasets were de-identified at their respective sites and subsequently transferred to Karolinska Institutet in an anonymized format. This study complies with the Helsinki Declaration. The patient sample collection was approved by the Stockholm Regional Ethics Committee (permits 2012/572-31/1, 2012/438-31/3, and 2018/845-32), the Swedish Ethical Review Authority (permit 2019-05220), and the Regional Committee for Medical and Health Research Ethics in Western Norway (permits REC/Vest 80924, REK 2017/71). Informed consent was obtained from patients in the Swedish dataset and was waived for other data cohorts due to the use of de-identified prostate specimens in a retrospective setting. Patient involvement in this study was supported by the Swedish Prostate Cancer Society.

## Results

### Interpretation of AI predictions and rationale for reduction in IHC use

We employed an in-house, task-specific AI model trained for prostate cancer grading[29] to assess its performance in retrospective cases requiring basal-cell IHC staining as part of routine clinical diagnostics. A prediction of "positive" would in this setting translate to "IHC-analysis recommended", indicating that the model is not confident that the WSI is benign relative to the applied threshold. Conversely, a "negative" prediction should be interpreted as "IHC analysis not recommended", indicating high AI confidence in benign morphology (i.e., a high negative predictive value) even at a sensitivity-prioritized threshold. In a scenario where the pathologist would have absolute trust in the thresholded AI predictions, i.e., only ordering IHC on positive-predicted WSIs, the amount of negative-predicted WSIs would represent IHC investigations saved compared to current diagnostic practice (Fig. 1). We evaluated the model by measuring sensitivity and specificity for prostate cancer detection at different sensitivity thresholds, along with the area under the receiver operating characteristic curve (AUC) (Fig. 2). Diagnostic performance was assessed across three validation cohorts (Table 1) representing only slides where pathologists ordered IHC-staining for basal-cell markers during routine diagnostics. These cohorts included WSIs from Stavanger University Hospital, Norway (SUH, n = 234 WSIs), Synlab Laboratory, France (SFR, n = 112 WSIs), and Synlab Laboratory, Switzerland (SCH, n = 164 WSIs).

With respect to the patient population, laboratory, and whole-slide scanner used for the digitization of biopsies, the SUH cohort represents an internal validation set (different patients but from the same scanner and lab as the AI training data), while the SFR and SCH cohorts represent entirely external validation sets (different patients, scanners, and laboratories than the training data). A detailed description of the data cohorts is provided in the predefined study protocol[30]. The AI model's performance was evaluated across varying sensitivity-prioritized thresholds for cancer probability (Table 2). In addition, we evaluated the performance of two foundation

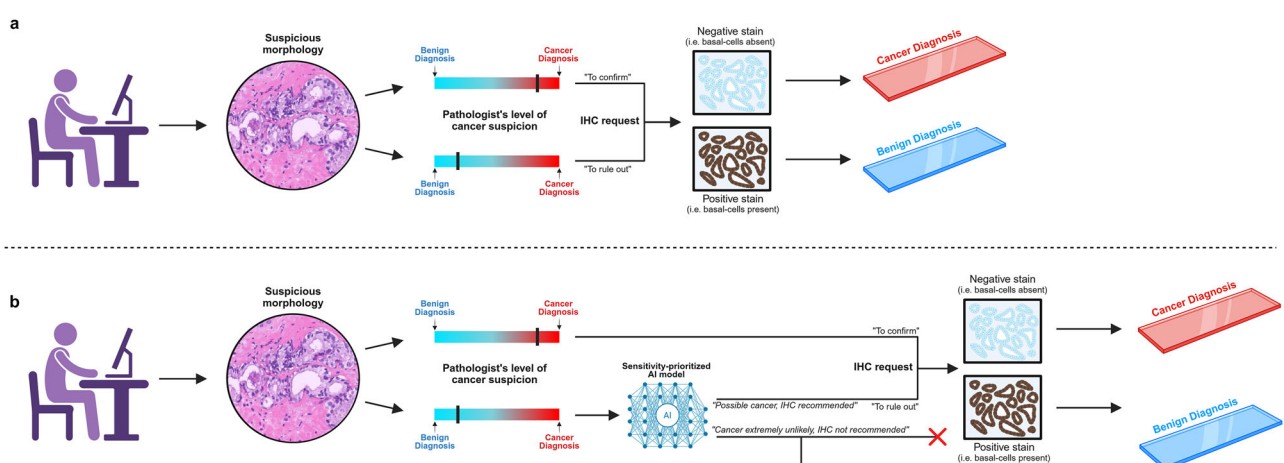

**Fig. 1 | Integration of the AI model into the diagnostic workflow.** In the current workflow (**a**), basal-cell-targeted immunohistochemistry (IHC) is a key tool for pathologists when encountering morphologies on H&E staining that cannot definitively be classified as benign or malignant. The degree of cancer suspicion varies across cases and between pathologists, and IHC is often requested even when suspicion is low. This practice contributes to increased workload, higher costs, and diagnostic delays. In the proposed workflow (**b**), the integration of a sensitivity-prioritized AI model, validated for accurately classifying ambiguous morphologies, aims to provide pathologists with additional assurance for diagnosing low-suspicion cases as benign without requiring IHC. The model assists by advising whether IHC is warranted when cancer suspicion arises; at a threshold of 0.01, cases with an AI-derived cancer probability exceeding 1% prompt a message recommending IHC. Conversely, when the cancer probability is extremely low (<1%), the AI advises against IHC and supports a benign diagnosis. In cases of low pathologist suspicion, agreement with the AI model may suffice for a benign diagnosis without further investigation. When a pathologist's suspicion is high, IHC is likely to be requested regardless of AI input. Figure created with BioRender (Blilie, A., 2025, https://BioRender.com/e23t809). AI artificial intelligence, H&E hematoxylin & eosin, IHC immunohistochemistry.

models: UNI (UFM) and Virchow2 (VFM) in this task (Supplementary Table 1).

## Diagnostic performance: internal validation
For the SUH internal validation cohort, the AI model achieved an AUC of 0.980 on IHC-validated WSIs. At the baseline threshold of 0.50, sensitivity was 0.914 and specificity was 0.930, yielding 120 true negatives and 9 false negatives out of 234 WSIs. Therefore, if IHC staining had only been ordered for positive AI labels, this threshold would have saved IHC for 129 out of 234 slides (55.0%), though 9 out of 105 cancer slides (8.6%) would have been missed. Using a highly sensitive threshold of 0.01 improved sensitivity to 1.0 while specificity dropped to 0.806, resulting in 104 true negatives and no false negatives. This adjustment would have saved IHC for 104 out of 234 slides (44.4%) without missing any cancers.

## Diagnostic performance: external validation
For the SFR external validation cohort, the model demonstrated an AUC of 0.993. At the 0.50 threshold, sensitivity was 0.935 and specificity was 0.955, with 63 true negatives and 3 false negatives among 112 WSIs. This would have saved IHC for 66 out of 112 slides (58.9%) while missing 3 out of 46 cancers (6.5%). Lowering the threshold to 0.4 eliminated all false negatives without losing any true negatives, resulting in 63 out of 112 IHC stains saved (56.3%). At the most sensitivity-prioritized threshold of 0.01, true negatives decreased to 47, reducing IHC savings to 47 out of 112 slides (42.0%).

For the SCH external validation cohort, the model achieved an AUC of 0.951. At a threshold of 0.50, sensitivity was 0.921 and specificity was 0.831, with 54 true negatives and 10 false negatives among 164 WSIs. This would have saved IHC for 64 out of 164 slides (39.0%) but missed 10 out of 99 cancers (10.1%). Using the highly sensitive-prioritized threshold of 0.01 increased sensitivity to 1.0, while specificity dropped to 0.523, resulting in 34 true negatives and reducing IHC savings to 34 out of 164 slides (20.7%).

## Pathologist review of false negative cases
At the unadjusted threshold of 0.50, false negative predictions were observed for a total of 22 WSIs across the SUH (9), SFR (3), and SCH (10) cohorts. Slide-level label data for ISUP grade and cancer length were available for SUH and SFR but not for SCH. In the SUH cohort, the nine false negatives had a mean cancer length of 2.8 mm (median: 1 mm, range: 0.2–11.0 mm) with the following ISUP distribution: ISUP 1: six slides, ISUP 4: one slide, and ISUP 5: two slides. For the SFR cohort, all three false negatives were ISUP 1 slides with cancer lengths of 2 mm, 4 mm, and 4 mm.

All 22 false negative WSIs were re-evaluated by the study pathologist (A.B.) in a blinded review. To maintain blinding, 12 additional external slides with a balanced distribution of all ISUP grades were included–the purpose of adding these slides was not to balance the dataset, but to mask the fact that the original cases were exclusively IHC-validated false negatives. The pathologist assessed only H&E-stained WSIs, providing a diagnosis for each case and indicating whether IHC would be required in a clinical setting. One WSI was diagnosed as benign, with no need for IHC. Sixteen WSIs were assigned non-definitive diagnoses of atypia (of uncertain significance) or suspicious for cancer (SFC), with IHC recommended for all. One WSI was diagnosed as ISUP 1 (3 + 3) cancer, which did not require IHC. Three WSIs were identified as high-grade cancers, including one ISUP 4 (4 + 4) and two ISUP 5 (5 + 5 and 5 + 4). Additionally, one WSI was deemed suspicious for ductal adenocarcinoma, necessitating IHC for a definitive diagnosis. Overall, IHC was recommended for 20 of the 22 cases.

Following this reassessment, a second pathologist (L.E.) participated in a review meeting where WSIs of IHC-stained slides, when available (SUH cohort), were presented alongside the WSIs of H&E-stained slides. L.E. is an experienced uropathology specialist and has been shown to be highly concordant with other specialists in earlier studies[6,18]. The consensus was that most false negative WSIs contained only minimal foci with ambiguous morphology, indeed warranting further IHC investigation. After evaluating the IHC stains, the pathologists agreed that in the majority of cases, the findings still did not meet the qualitative and quantitative criteria for a definitive cancer diagnosis (Fig. 3).

In 18 of the 22 cases, the suspicious areas displayed low-grade morphology, with at most minimal ISUP 1 cancer. One case had a consensus diagnosis of probable ductal adenocarcinoma. The remaining three cases, classified as high-grade cancers, were also independently assessed by the second pathologist (L.E.) in a blinded review. Both pathologists confirmed

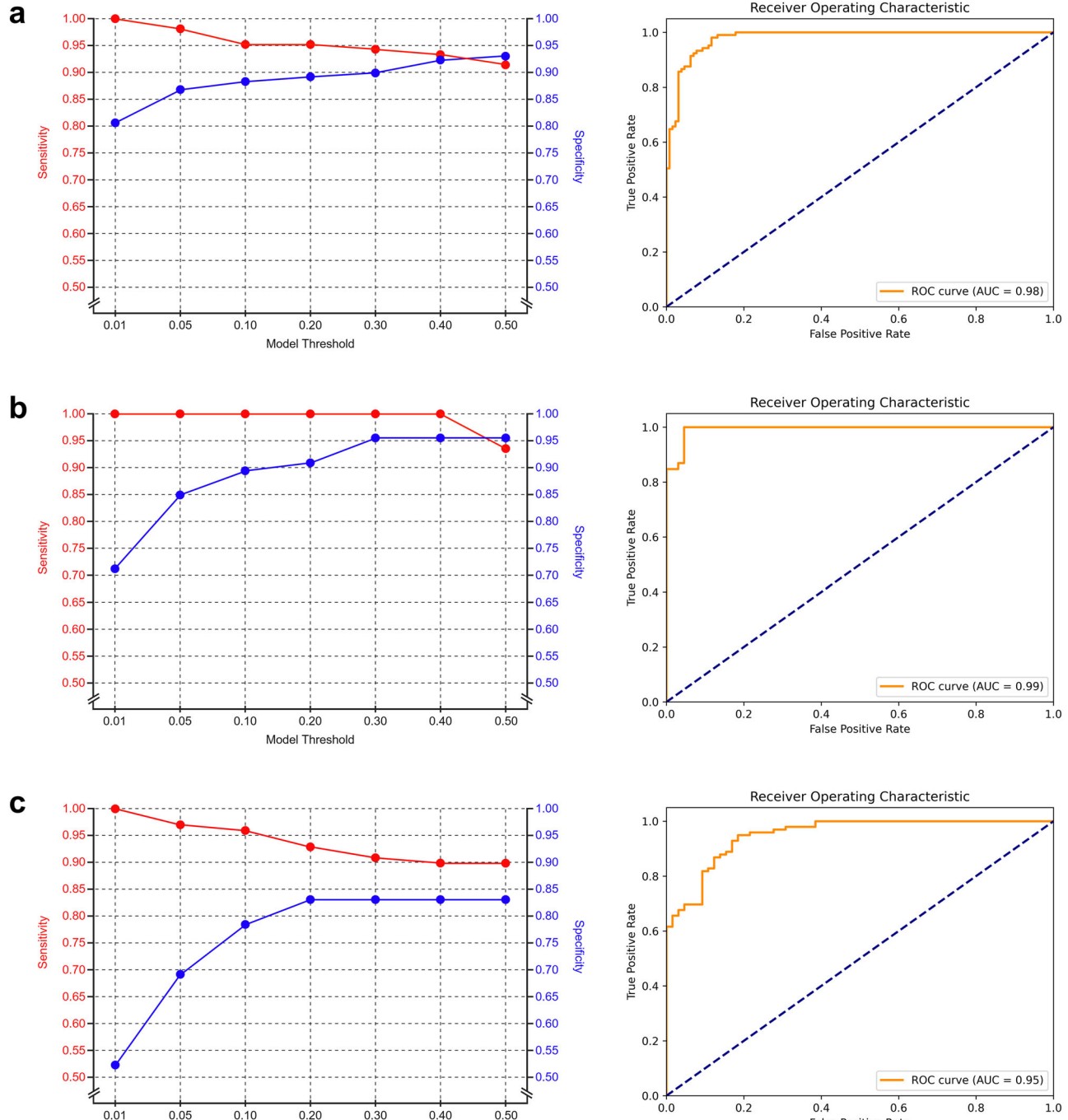

**Fig. 2 | Model performance across three cohorts using sensitivity-prioritized thresholding.** Sensitivity/specificity (left) and ROC (right) curves for the SUH (**a**), SFR (**b**), and SCH (**c**) cohorts. The sensitivity/specificity curves illustrate the trade-off between sensitivity and specificity at various model thresholds. We apply the model at a low threshold to maximize sensitivity while maintaining sufficiently high specificity to significantly reduce unnecessary IHC staining for true negative (non-cancerous) cases. The ROC curves illustrate the model's overall discrimination capacity, as summarized by the AUC metric. AUC area under the curve, IHC immunohistochemistry, ROC receiver operating characteristic, SCH Synlab Laboratory Switzerland, SFR Synlab Laboratory France, SUH Stavanger University Hospital.

ISUP grades consistent with the original reports. However, all three WSIs exhibited significant crush artifacts and tissue folds, partially obscuring cancer morphology. Notably, in these cases, the pathologists who made the original diagnoses had requested PSA immunostaining alongside basal-cell stains to confirm the prostatic origin of the malignancies. This finding aligns with the meeting consensus that these false negatives represented true high-grade cancers with atypical features for acinar carcinoma. Figure 4 provides representative images of the high-grade areas.

Importantly, the AI model can provide attention maps along with prostate cancer diagnosis predictions. Attention heatmaps for specifically these false negative slides reveal that, although the model ultimately predicted them as benign, the AI correctly localized and highlighted suspicious areas within the tissue. One pathologist (A.B.) independently reviewed all false-negative WSIs in full and annotated the regions deemed most suspicious for malignancy. These initial assessments were then discussed in a consensus meeting with a second pathologist (L.E.), during which the

annotated regions were compared to both the corresponding IHC staining patterns (where available) and the AI model's attention maps. This review confirmed a strong correspondence between the pathologist-identified areas of concern, IHC-confirmed regions, and the model's high-attention zones. This suggests that even when an AI model does not flag a case for an IHC order, the attention heatmaps could serve as an additional layer of decision support, helping pathologists focus on diagnostically challenging regions.

## Discussion

Our results show that the AI model retains high diagnostic performance even for morphologies deemed ambiguous by pathologists (i.e., slides where the pathologists required IHC to make the final diagnosis of benign vs. cancer). By thresholding predictions in a sensitivity-prioritized fashion, we demonstrate the potential of using AI as a decision-support system for deciding when IHC staining is truly necessary. Ordering IHC for every ambiguous WSI with a predicted cancer probability exceeding 1% (sensitivity-maximized threshold of 0.01) eliminated all false negatives (sensitivity = 1.0), while still significantly reducing IHC staining performed on benign slides (44.4%, 42.0%, and 20.7% total IHC reduction for cohorts SUH, SFR and SCH, respectively). The observed differences in IHC reduction across cohorts can be partially explained by cohort composition—specifically, the proportion of benign slides. The SUH and SFR cohorts included a higher percentage of benign cases (55.1% and 58.9%, respectively) compared to the SCH cohort (39.6%). Since our approach targets IHC savings exclusively for benign slides, a smaller overall impact in the SCH cohort is expected. However, even when considering only slides with a benign ground truth, the IHC savings vary substantially across sites—80.6% for SUH, 71.2% for SFR, and 52.3% for SCH—indicating that other factors contribute as well. We believe this variation likely reflects differences in institutional and individual IHC-ordering practices. For example, some sites may have a lower threshold for initiating IHC, applying it even for mildly suspicious morphologies, while others may reserve IHC for cases with more overt atypia. Such differences in diagnostic thresholds and practice patterns can meaningfully influence the potential for IHC reduction. The performance of the task-specific model was similar to FMs, supporting the general applicability of different AI models for this purpose. The FMs exhibited slightly higher sensitivity compared to the task-specific model but at the cost of lower specificity, consistent with previous findings[29].

Pathologists' reassessment of the false negative WSIs observed at higher thresholds revealed that the vast majority of these slides contained only minimal foci of low-grade morphologies, warranting diagnoses of atypia or SFC rather than definitive malignant classification. Such diagnoses are applied when the morphological features are insufficient for a conclusive cancer diagnosis, yet there remains some degree of uncertainty, and malignancy cannot be ruled out. This category of indeterminate diagnoses also includes "atypical small acinar proliferation (ASAP)", although the use of this term is discouraged by the International Society of Urological Pathology (ISUP)[36,37].

One of the false negative predictions involved a case of ductal adenocarcinoma. While not excluded from the datasets, this is a cancer subtype our AI model is not specifically trained or validated to detect. Although this is the second most common subtype of prostate cancer after acinar adenocarcinoma, it remains rare, comprising only 0.17% of cases[38]. Due to its low prevalence, acquiring sufficient training data for robust AI model development and validation remains a significant challenge. This case also highlights the broader issue of detecting and differentiating various intraductal proliferations such as high-grade prostatic intraepithelial neoplasia (HGPIN), atypical intraductal proliferation (AIP), and intraductal carcinoma (IDC). Our current model is not validated to distinguish these entities, and they fall outside the scope of this study. While IHC—particularly basal-cell markers—can aid in distinguishing IDC from invasive cribriform (Gleason pattern 4) or comedonecrotic (Gleason pattern 5) cancers, current guidelines recommend IHC primarily in cases lacking definitive invasive cancer, which are relatively rare (0.06–0.26% of biopsies)[39]. Moreover, the diagnostic value of basal-cell IHC in differentiating HGPIN, AIP, and IDC is

**Table 2 | AI model performance across sensitivity-prioritized thresholds**

| Cohorts | SUH | SFR | SCH |
|---|---|---|---|
| | $n$ = 234 | $n$ = 112 | $n$ = 164 |
| **AUROC** | **0.980** | **0.993** | **0.951** |
| Threshold 0.50 | | | |
| Sensitivity | 0.914 | 0.935 | 0.899 |
| Specificity | 0.930 | 0.955 | 0.831 |
| True positives (TP) | 96 | 43 | 89 |
| False positives (FP) | 9 | 3 | 11 |
| True negatives (TN) | 120 | 63 | 54 |
| False negatives (FN) | 9 (8.6%) | 3 (6.5%) | 10 (10.1%) |
| ISUP 1 | 6 | 3 | 6* |
| ISUP 2 | 0 | 0 | 1* |
| ISUP 3 | 0 | 0 | 3* |
| ISUP 4 | 1 | 0 | 0 |
| ISUP 5 | 2 | 0 | 0 |
| IHC reduction (TN + FN) | 129 (55.0%) | 66 (58.9%) | 64 (39.0%) |
| Threshold 0.20 | | | |
| Sensitivity | 0.952 | 1.000 | 0.929 |
| Specificity | 0.892 | 0.909 | 0.831 |
| True positives (TP) | 100 | 46 | 92 |
| False positives (FP) | 14 | 6 | 11 |
| True negatives (TN) | 115 | 60 | 54 |
| False negatives (FN) | 5 (4.8%) | 0 (0.0%) | 7 (7.1%) |
| ISUP 1 | 2 | 0 | 3* |
| ISUP 2 | 0 | 0 | 1* |
| ISUP 3 | 0 | 0 | 3* |
| ISUP 4 | 1 | 0 | 0 |
| ISUP 5 | 2 | 0 | 0 |
| IHC reduction (TN + FN) | 120 (51.3%) | 60 (53.6%) | 61 (37.2%) |
| Threshold 0.01 | | | |
| Sensitivity | 1.000 | 1.000 | 1.000 |
| Specificity | 0.806 | 0.712 | 0.523 |
| True positives (TP) | 105 | 46 | 99 |
| False positives (FP) | 25 | 19 | 31 |
| True negatives (TN) | 104 | 47 | 34 |
| False negatives (FN) | 0 (0.0%) | 0 (0.0%) | 0 (0.0%) |
| ISUP 1 | 0 | 0 | 0 |
| ISUP 2 | 0 | 0 | 0 |
| ISUP 3 | 0 | 0 | 0 |
| ISUP 4 | 0 | 0 | 0 |
| ISUP 5 | 0 | 0 | 0 |
| IHC reduction (TN + FN) | 104 (44.4%) | 47 (42.0%) | 34 (20.7%) |

AI model performance across the SUH, SFR, and SCH cohorts under different sensitivity-prioritized thresholds. In a scenario where IHC staining is only requested for AI-predicted positive slides, the reduction in IHC usage corresponds to the total number of negative predictions. False negative predictions indicate missed cancers, and their ISUP distribution is provided. *In the SCH cohort, cancer grading was assigned at the location level (across multiple slides), meaning true grades for individual WSIs are unknown. Instead, we report the overall grade assigned to the location. *AI* artificial intelligence, *AUROC* area under the receiver operating characteristic curve, *IHC* immunohistochemistry, *ISUP* International Society of Urological Pathology grade, *SCH* Synlab Laboratory Switzerland, *SFR* Synlab Laboratory France, *SUH* Stavanger University Hospital.

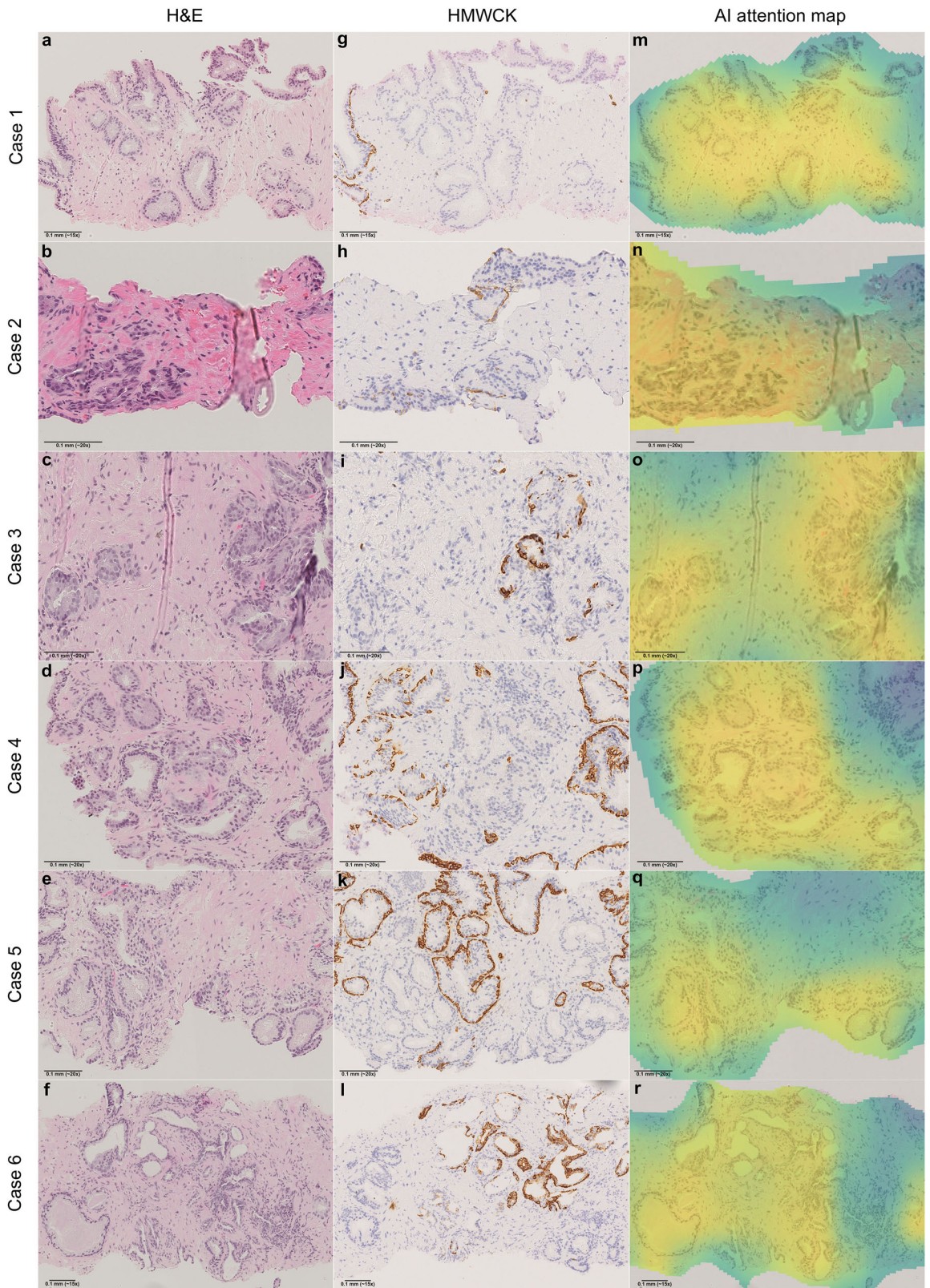

H&E — HMWCK — AI attention map; Case 1 through Case 6 (panels a–r).

limited; such assessments continue to rely heavily on expert interpretation of H&E morphology, occasionally supplemented with non-basal-cell markers such as AMACR. As we do not currently report separate performance metrics for ductal adenocarcinoma or intraductal lesions, these limitations further underscore the importance of human oversight and the current role of AI as a decision-support tool rather than a stand-alone diagnostic system.

Nevertheless, we aim to expand our dataset to include more such cases in future iterations of the model.

The three false negative cases representing high-grade cancers (one ISUP 4 and two ISUP 5) were confirmed as such by both study pathologists during reassessment, consistent with the original reports. Importantly, the infiltrative nature of these lesions was readily apparent, making it highly

**Fig. 3 | False negative predictions with low-grade morphologies from the SUH cohort.** At the base model threshold of 0.5, i.e., before sensitivity prioritization, the AI produced six false negative predictions exhibiting low-grade morphologies from the SUH cohort–the only cohort with digitized IHC stains available. These cases were reassessed by the study pathologist (A.B.), blinded to the AI result, and subsequently reviewed in a meeting with a second pathologist (L.E.). Case 1 (**a, g**) shows a sub-millimeter area of prostatic glands with enlarged nuclei and prominent nucleoli, suspicious for ISUP 1 cancer. However, features like a fuzzy luminal border, wavy contours, and cytoplasmic pigment make a definitive diagnosis difficult. IHC confirmed a small ISUP 1 cancer; otherwise, the case would have been diagnosed as atypia/SFC. Case 2 (**b, h**) shows small, angulated glands and stromal elastoid degeneration, suggesting postatrophic hyperplasia. Some nuclear irregularities and hyperchromasia are present, but not convincing of malignancy. IHC reveals partially retained basal cells. The pathologists would render a benign diagnosis. Case 3 (**c, i**) shows 3-4 glands (left) with nuclear enlargement and prominent nucleoli, highly suspicious for malignancy. IHC confirmed ISUP 1 cancer, though the quantity is borderline insufficient for a definitive diagnosis. Case 4 (**d, j**) shows approximately 10 glands with low-grade atypia, not convincing for malignancy. On IHC, basal cells are lost in some glands; however, other glands with similar H&E morphology retain them. The pathologists would diagnose this as atypia/SFC, even after IHC. Case 5 (**e, k**) shows glands with minimal nuclear atypia deemed insufficient for a definitive cancer diagnosis, regardless of the IHC result. The pathologists would diagnose this as atypia/SFC. Case 6 (**f, l**) shows two glands with obvious nuclear atypia and pathological secretion, but quantitatively, the suspicious glands are too few to render a definitive malignant diagnosis, even after IHC. The pathologists would diagnose atypia/SFC, noting a strong suspicion of ISUP 1 cancer. AI-generated attention maps (**m–r**) show that, although slides were ultimately predicted to be benign, the AI correctly identifies suspicious areas, which could help focus the pathologist's attention in a clinical setting. AI artificial intelligence, H&E hematoxylin & eosin, HMWCK high-molecular-weight cytokeratin, IHC immunohistochemistry, ISUP International Society of Urological Pathology grade, SFC suspicious for cancer, SUH Stavanger University Hospital.

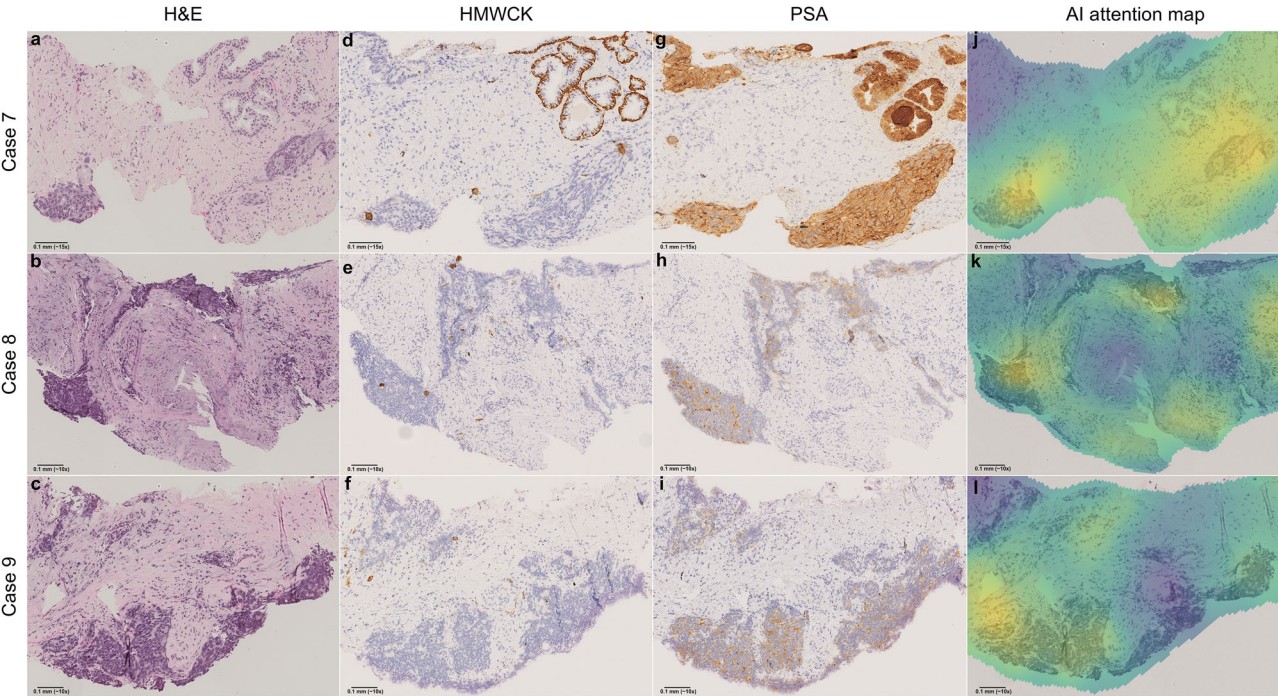

**Fig. 4 | False negative predictions with high-grade morphologies from the SUH cohort.** At the base model threshold of 0.5, i.e., prior to sensitivity prioritization, the AI made three false negative predictions for slides showing high-grade morphologies from the SUH cohort. These cases were reassessed by both study pathologists (A.B., L.E.), blinded to the AI result and to each other, and later discussed in a review meeting. Case 7 (**a, d, g**) shows two groups of fused prostatic glands with enlarged nuclei and prominent nucleoli, distinct from the benign glands seen at the top of the image. The cells are markedly atypical with an infiltrative appearance, though morphology is somewhat obscured by crush artifacts. Subtle luminal openings support a diagnosis of ISUP 4 cancer (GS 4 + 4), which was the diagnosis of both study pathologists during the blinded assessment. This 1 mm focus was the only malignant area in the biopsy. IHC shows basal-cell loss, confirming infiltration, and PSA expression confirms the prostatic origin of the malignancy. Case 8 (**b, e, h**) shows a focus of infiltrating prostatic carcinoma with a solid architecture, correctly diagnosed as ISUP 5 cancer (GS 5 + 5) by both pathologists during blinded assessment. Tissue folds and severe crush artifacts obscure morphology, but the malignant nature of the tissue is unquestionable. Apoptosis and nuclear molding raise the possibility of large-cell neuroendocrine malignancy. Multiple small tumor foci were present in the biopsy core. IHC confirms infiltration through basal-cell loss, and a positive PSA stain, however weak, rules out the differential diagnosis of neuroendocrine carcinoma. Case 9 (**c, f, i**) represents a biopsy from the same patient as Case 8, showing similar morphology and IHC expression. Despite areas of significant crush artifacts in the tumor tissue, the high-grade malignant nature of the tumor is obvious. ISUP 5. Heatmaps (**j–l**) generated by the AI model partly highlight the malignant areas, directing the attention of the pathologist toward these regions in a clinical setting. However, for these high-grade cases, we believe the likelihood of a pathologist missing them during routine diagnostics is minimal. Note that all slides were correctly classified as malignant when sensitivity-prioritized thresholding was applied. AI artificial intelligence, GS Gleason Score, H&E hematoxylin & eosin, HMWCK high-molecular-weight cytokeratin, IHC immunohistochemistry, ISUP International Society of Urological Pathology grade, PSA prostate-specific antigen, SUH Stavanger University Hospital.

unlikely for these cancers to be missed in clinical practice. This emphasizes the role of AI models as diagnostic aids for pathologists, with the final decision remaining under human oversight. It is also worth noting that while the evaluation presented in this study was conducted on individual slides, several slides are typically assessed per prostate. As multiple WSIs are screened per patient, the probability of a false-negative cancer diagnosis is further reduced. Still, a pathologist's assessment remains crucial to ensure accurate diagnoses when encountering technical artifacts or rare morphological variants that the AI model has not been sufficiently exposed to during development[40]. While stain variation and tissue artifacts can affect AI performance, our primary validation study[29] demonstrated model robustness against stain variation across multiple fully external datasets. Additionally,

several color calibration techniques have been proposed to further enhance model robustness to staining variability[41]. In contrast, tissue artifacts that obscure key morphological features remain a significant challenge. We have previously investigated this issue and plan to implement a conformal prediction framework to flag uncertain predictions—such as those arising from obscured or artifact-laden tissue—in future versions of the model[40]. Regarding thresholding, the review process highlights the importance of prioritizing sensitivity. Although many of the false negative WSIs encountered at threshold 0.5 should likely be diagnosed as atypia/SFC rather than definitive malignancy, their need for IHC staining suggests that the baseline threshold of 0.5 is insufficient for this use case.

A critical factor influencing the adoption of AI in pathology is whether pathologists will trust its predictions. Our proposed scenario, where the AI model would suggest omitting IHC for morphologies the pathologist perceives as ambiguous, is no exception–the thought of misclassifying cancer as benign due to AI advice is naturally frightening. However, we must understand that the pathologists' uncertainty spans a continuous spectrum. Sometimes the cancer suspicion is very low, but using IHC validation as a safety net is an easy way of eliminating lingering doubt. The fear of missing malignancies, combined with the fact that most doctors are not involved in departmental financial governance, could explain overly cautious approaches, where pathologists prioritize ensuring accurate diagnoses over the institution's financial considerations. This tendency is reflected in our data with 55.1%, 58.9%, and 39.6% of IHC-validated WSIs ultimately yielding benign diagnoses from the SUH, SFR, and SCH cohorts, respectively. We believe that for cases where initial cancer suspicion is low, the addition of an AI model proven to be highly adept at correctly classifying cancer vs. benign tissue in similar situations could give pathologists the extra assurance needed to sign out benign samples without IHC validation. The potential impact of reducing IHC usage depends on institutional practices, which vary. However, our data suggest significant potential for reduction: IHC was requested for 56%, 58%, and 38% of patients in the SUH, SFR, and SCH cohorts, respectively, and for 20%, 22%, and 7% of all slides in those same cohorts. In the likely situation of early resistance from pathologists, trust could develop over time as they use the AI model and observe its consistent accuracy. Confidence may be gained by pathologists initially sticking to their individual IHC-ordering patterns while cross-verifying AI predictions with subsequent IHC results. This trust-building process would be essential for encouraging widespread acceptance and integration of AI in routine diagnostics.

To date, there are very few publications focusing on the utilization of AI models in IHC-related tasks within prostate pathology. A study by Chatrian et al.[16] aimed to pre-order IHC for presumed difficult cases in order to save diagnostic time for the investigating pathologist, using an AI model trained on cases from routine diagnostics where IHC staining had been performed. That is, the aim of the AI model was to mimic the IHC requesting pattern of pathologists. Our work is fundamentally different as we aim to provide pathologists with an AI model that will modify these patterns, reducing the number of IHCs requested for truly benign cases where pathologists' suspicion of cancer is low. While the approach of Chatrian et al. enhances workflow efficiency, it does not address the overuse of IHC in benign cases, which represents a tangible resource burden. By using a sensitivity-prioritized thresholding framework, our AI model offers a novel solution to this issue, allowing pathologists to confidently forgo IHC in benign cases while maintaining diagnostic accuracy for malignant cases.

Eloy et al. demonstrated how using an AI model in the evaluation of prostate biopsies reduced the reliance on IHC workup compared to the traditional diagnostic approach[27]. The study design involved four pathologists assessing the same set of slides in two phases, with a washout period of a minimum of 2 weeks between assessments. All slides in the set were presumably difficult cases, having had IHC requested during the routine diagnostic process. Phase 1 involved assessment with no aid from AI, while in Phase 2, the AI model was introduced. Even though the results showed a reduction of pathologist IHC requests when slide assessment was assisted by AI, there are reasons to question the relevance and transferability of these findings to routine diagnostic practice. Firstly, the pathologists were aware that all slides in the set had IHC staining performed during primary diagnostics. Secondly, Phase 1 allowed pathologists to view IHC stains, given that they would have requested it in a diagnostic situation. Knowing that other pathologists had requested IHC, and having these stains readily available in a research setting without real-world consequences in terms of time or resources if choosing to look at them, risks introducing bias. Furthermore, giving the pathologists the option of seeing the IHC stains in Phase 1, i.e., letting them know the true nature of the tissue, could potentially have introduced bias in Phase 2.

Our study highlights the potential of a sensitivity-prioritized AI framework for reducing IHC use for benign prostate biopsies, alleviating resource burdens, reducing costs, and improving diagnostic efficiency in pathology laboratories. The AI model demonstrates state-of-the-art performance, maintaining high sensitivity and specificity even in challenging cases where pathologists traditionally rely on IHC. In a significant proportion of these cases, the AI model shows overwhelming confidence in its predictions, underscoring its potential to reduce IHC staining for benign slides even when thresholds are applied. However, it must be noted that the present analysis is limited to IHC data from three cohorts. While our previous large-scale validation study demonstrated that the AI model generalizes well across 12 external cohorts from 11 countries, those evaluations did not specifically focus on diagnostically challenging cases requiring IHC. This highlights the need for further validation in broader and more diverse clinical settings, particularly for difficult cases. By standardizing decision-making across pathologists with varying experience levels, AI has the potential to mitigate subjectivity in IHC usage and enhance diagnostic consistency.

Integration of AI into clinical workflows requires careful consideration of laboratory protocols, workflow dynamics, and user interactions, and prospective studies in real-world settings will be crucial for validating the clinical and economic benefits suggested by our retrospective analysis. While our study focuses on diagnostic performance and potential IHC savings, broader implementation of AI in pathology will also require careful consideration of integration costs. This includes infrastructure requirements, image processing times, IT maintenance, and long-term operating expenses. We advocate for future cost-effectiveness studies that comprehensively evaluate these factors alongside diagnostic benefits. Such analyses will be critical in assessing the real-world value of AI-assisted pathology workflows, and we plan to pursue this in collaboration with health economics experts. Ultimately, AI-driven pathology represents a transformative opportunity to improve diagnostic precision, streamline workflows, and optimize resource utilization, contributing to better patient outcomes worldwide.

## Data availability

All relevant data are available upon request, but cannot be shared publicly. For any requests to access these sources, inquiries should be directed to M.E. at Karolinska Institutet (martin.eklund@ki.se). Requests will be evaluated on a case-by-case basis, with approvals granted if they comply with data privacy regulations and intellectual property policies. A subset of the data used for model training (STHLM3 and RUMC cohorts) is available for non-commercial purposes, subject to a CC BY-SA-NC 4.0 license as part of the PANDA challenge dataset and is freely downloadable after registration at https://www.kaggle.com/c/prostate-cancer-grade-assessment. Source data are provided with this paper (Supplementary Data Set 1).

## Code availability

No significant custom code was developed for this study. The AI models were implemented as described in the previous study[29]. Torch library (https://github.com/pytorch/pytorch) was used for obtaining model predictions. For the foundation models, we have used the publicly available models from https://huggingface.co/paige-ai/Virchow2 and https://huggingface.co/MahmoodLab/UNI.

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

## Acknowledgements

A.B. received a grant from the Health Faculty at the University of Stavanger, Norway. M.E. received funding from the Swedish Research Council, Swedish Cancer Society, Swedish Prostate Cancer Society, Nordic Cancer Union, Karolinska Institutet, and Region Stockholm. K.K. received funding from the SciLifeLab & Wallenberg Data Driven Life Science Program (KAW 2024.0159), David and Astrid Hägelen Foundation, Instrumentarium Science Foundation, KAUTE Foundation, Karolinska Institute Research Foundation, Orion Research Foundation, and Oskar Huttunen Foundation. We thank Silja Kavlie Fykse and Desmond Mfua Abono for scanning in Stavanger. We would like to acknowledge the patients who contributed the clinical information that made this study possible. Computations were enabled by the National Academic Infrastructure for Supercomputing in Sweden (NAISS) and the Swedish National Infrastructure for Computing (SNIC) at C3SE, partially funded by the Swedish Research Council through grant agreement no. 2022-06725 and no. 2018-05973, and by the supercomputing resource Berzelius provided by the National Supercomputer Centre at Linköping University and the Knut and Alice Wallenberg Foundation.

## Author contributions

A.B., M.T., G.M.G., J.A., M.G., P.L., E.G., S.R.K., and E.A.M.J. collected, assessed, and curated clinical datasets. A.B., N.M., X.J., K.S., S.E.B., M.T., and K.K. contributed to the digitization, pre-processing, and management of WSI data. N.M., X.J., K.S., S.E.B., and K.K. developed the AI models. A.B. and N.M. conducted the statistical analyses. A.B. and L.E. conducted an in-depth review of false negative cases. A.B., N.M., X.J., S.E.B., L.E., E.A.M.J., M.E., and K.K. analyzed and interpreted the study results. A.B., M.E., and K.K. acquired funding. K.K. conceived of the study and takes responsibility for its integrity and accuracy. A.B., N.M., M.E., and K.K. drafted the manuscript. All authors reviewed, edited, and approved the manuscript.

## Funding

## Competing interests

N.M., L.E., K.K., and M.E. are shareholders of Clinsight AB. All other authors declare no competing interests.

## Additional information

[1]Department of Pathology, Stavanger University Hospital, Stavanger, Norway. [2]Faculty of Health Sciences, University of Stavanger, Stavanger, Norway. [3]Department of Medical Epidemiology and Biostatistics, Karolinska Institutet, Stockholm, Sweden. [4]Department of Molecular Medicine and Surgery, Karolinska Institutet, Stockholm, Sweden. [5]Department of Pathology, Synlab, Madrid, Spain. [6]Department of Pathology, Synlab, Brescia, Italy. [7]The General Practice and Care Coordination Research Group, Stavanger University Hospital, Stavanger, Norway. [8]Department of Global Public Health and Primary Care, Faculty of Medicine, University of Bergen, Bergen, Norway. [9]Department of Oncology and Pathology, Karolinska Institutet, Stockholm, Sweden. [10]Faculty of Science and Technology, University of Stavanger, Stavanger, Norway. [11]Institute for Biomedicine and Glycomics, Griffith University, Queensland, Australia. [12]Department of Medical Epidemiology and Biostatistics, SciLifeLab, Karolinska Institutet, Stockholm, Sweden. [13]These authors contributed equally: Anders Blilie, Nita Mulliqi. ✉e-mail: kimmo.kartasalo@ki.se

