## [Transparent Peer Review file · Communications Medicine]

Artificial Intelligence-Assisted Prostate Cancer Diagnosis for Reduced Use of Immunohistochemistry

Corresponding Author: Dr Kimmo Kartasalo

Version 0:

Reviewer comments:

Reviewer #1

(Remarks to the Author)

This study demonstrates that an AI model can maintain high diagnostic accuracy even in ambiguous prostate biopsy cases where pathologists typically rely on immunohistochemistry (IHC) for confirmation. By applying a sensitivity-prioritized threshold (e.g., 1% cancer probability), the AI system effectively identifies cases where IHC is truly necessary. This approach eliminates false negatives while significantly reducing unnecessary IHC use—by up to 44.4% in benign cases across multiple cohorts.

Importantly, the few false negatives identified were primarily minimal or low-grade lesions that did not meet criteria for definitive cancer diagnoses. While IHC usage varies by institution, the findings suggest that AI integration could lead to substantial cost savings and improved workflow efficiency. The authors emphasize that AI should serve as a supportive tool rather than a replacement for pathologists, and that trust and gradual integration are essential for successful adoption.

Comment 1:

The AI model operates on whole-slide images (WSIs) of H&E-stained prostate biopsies, outputting a probability score that reflects the likelihood of cancer presence in specific tissue regions. This model shows promise for enhancing diagnostic workflows and reducing resource burdens.

However, several important questions remain. One notable false negative involved a diagnosis of ductal adenocarcinoma, raising concerns about the model's ability to detect the spectrum of intraductal lesions, including high-grade prostatic intraepithelial neoplasia (HGPIN), atypical intraductal proliferation (AIP), and intraductal carcinoma (IDC) [an area in which IHC can often be helpful]. Clarification is needed on how the model handles these entities within its probability scoring framework.

Comment 2:

Additionally, the study acknowledges that tissue artifacts and stain variation can impact model accuracy. It would be valuable to know whether the authors anticipate improvements in the model's robustness to such challenges, and whether future iterations will incorporate mechanisms to better handle these limitations.

In conclusion, this AI model represents a compelling advancement in prostate pathology, with the potential to enhance diagnostic efficiency and reduce unnecessary IHC use. It could become a valuable tool for practicing pathologists, supporting more cost-effective and streamlined workflows.

Reviewer #2

(Remarks to the Author)

The work uses an AI model on H&E images of prostate core needle biopsies from three different cohorts to decreased the need for immunohistochemistry.

A well-written, concise, and direct manuscript with impact in the healthcare management.

The work could benefit from minor revisions/clarifications, listed below:

- It is not stated whether the immunohistochemistry technique developed at the three sites is the same (protocol, antibody

used – namely the clone, equipment, etc.); the same for the HE staining

- Details of scanning and navigation are missing, like magnification, and how this data might interfere with both visual and computational analysis.
- It would be interesting to try and explain the difference in IHC request reduction across the cohorts based on their origin: with two being close to 40% (SUF and SFR) and one around 20% (SCH). How do you explain this? What factors might have influenced it?
- The review of cases by pathologists is done with the addition of 12 cases for balancing, considering the various ISUP groups – how is this distribution done?
- With a threshold of 0.50, 22 false negative WSIs were re-evaluated. How do you interpret the impact of technical quality and control, given that the three high-grade cancer false negative cases presented artifacts?
- It is stated, without additional data, that pathologists confirmed the AI model identified areas of interest – it was not explained how this evaluation was conducted.
- It seems risky to state that since there are multiple WSIs from the same patient, the probability of false negatives in cancer diagnosis is reduced.
- Regarding Eloy C's article, it mentions that the 2-week wash-out period is short, but without scientific evidence (the College of American Pathologists uses this period as a reference for WSI validation in a clinical context).

More limitations of the work should be mentioned, rather than just the limitations of other studies, such as: low variability in the training set (which will be a difficulty for implementation in other laboratory settings); lack of information regarding laboratorial procedures and eventual variability which can impact the AI model performance; the image processing time, IT infrastructure, integration methods, and maintenance costs for using this AI model were not mentioned.

Reviewer #3

(Remarks to the Author)

This is an interesting study that could potentially have significant impact on anatomic pathology laboratory workflow and cost reduction. It provides preliminary findings that are promising. One minor comment I have is that some of the statements that are being made may be on the side of overstatements as the data need further validation (more slides from more institutions, different practice settings, etc.). Additionally, authors should consider discussing whether/how the implementation of such AI in the real-life practice setting will be actually reducing the overall costs (i.e. will the benefit of the AI implementation (reducing # of IHCs) outweigh the cost of AI implementation).

Version 1:

Reviewer comments:

Reviewer #1

(Remarks to the Author)

Thank you for your additional comments and explanations. Concerns have been adequately addressed.

Reviewer #2

(Remarks to the Author)

No additional comments for author.

Reviewer #3

(Remarks to the Author)

Thoughtful rebuttal letters and all the reviewer points seem to have been addressed and answered. No further comments from my end. Looking forward to seeing this project potentially make an impact in our field.

Response to reviewers (COMMSMED-25-0850-T)

We thank you for the assessment and rapid feedback on our manuscript. We have revised the manuscript accordingly and provide our response below (original comments in black, our response in blue).

In addition to revisions based on reviewer comments, the following error has been corrected in the Results (line 135):

At the 0.50 threshold, sensitivity was 0.935 and specificity was 0.955, with 63 true negatives and 3 false negatives among 112 WSIs. This would have saved IHC for ~~63~~ 66 out of 112 slides (58.9%) while missing 3 out of 46 cancers (6.5%).

Reviewer #1 (Remarks to the Author):

This study demonstrates that an AI model can maintain high diagnostic accuracy even in ambiguous prostate biopsy cases where pathologists typically rely on immunohistochemistry (IHC) for confirmation. By applying a sensitivity-prioritized threshold (e.g., 1% cancer probability), the AI system effectively identifies cases where IHC is truly necessary. This approach eliminates false negatives while significantly reducing unnecessary IHC use—by up to 44.4% in benign cases across multiple cohorts.

Importantly, the few false negatives identified were primarily minimal or low-grade lesions that did not meet criteria for definitive cancer diagnoses. While IHC usage varies by institution, the findings suggest that AI integration could lead to substantial cost savings and improved workflow efficiency. The authors emphasize that AI should serve as a supportive tool rather than a replacement for pathologists, and that trust and gradual integration are essential for successful adoption.

We thank the reviewer for their summary and positive comments on the study. We would also like to point out that the reduction of 44.4% quoted by the reviewer is in fact the total reduction of IHC use for all slides. For benign cases, the reduction is actually up to 80.6%.

Comment 1:

The AI model operates on whole-slide images (WSIs) of H&E-stained prostate biopsies, outputting a probability score that reflects the likelihood of cancer presence in specific tissue regions. This model shows promise for enhancing diagnostic workflows and reducing resource burdens.

However, several important questions remain. One notable false negative involved a diagnosis of ductal adenocarcinoma, raising concerns about the model's ability to detect the spectrum of intraductal lesions, including high-grade prostatic intraepithelial neoplasia (HGPIN), atypical intraductal proliferation (AIP), and intraductal carcinoma (IDC) [an area in which IHC can often be helpful]. Clarification is needed on how the model handles these entities within its probability scoring framework.

While ductal adenocarcinoma is the second most common subtype of prostate cancer after acinar adenocarcinoma, it still only accounts for 0.17% of prostate cancers (<https://doi.org/10.1002/bco2.60>). Thus, it is challenging to acquire substantial amounts of training data for development of an AI model to reliably detect this cancer subtype. Our model has not been specifically evaluated for the diagnosis of ductal adenocarcinoma, but we are aiming to collect additional data for validating the model on ductal adenocarcinoma samples in the future.

Regarding the various intraductal proliferations encountered in prostate biopsies, our model is not specifically validated for differentiating between these lesions. While it is true that IHC can be helpful in distinguishing IDC from invasive cribriform (GP4) or comedonecrotic (GP5) cancer, current guidelines only recommend this approach in cases where there is no evidence of invasive cancer alongside the possible IDC areas—which is rare: 0.06-0.26% of biopsies (<https://doi:10.1097/PAS.0000000000001497>). The value of basal-cell IHC stains in distinguishing between different intraductal proliferations (HGPIN, AIP, IDC) is limited. The diagnosis in these cases primarily relies on careful evaluation of H&E morphology, occasionally supplemented by non-basal-cell IHC markers such as AMACR.

Currently, we do not have results on model performance for ductal adenocarcinoma and intraductal lesions. This further highlights the need for human assessment, and that the AI should not be used as a stand-alone tool for prostate cancer diagnosis at the current stage.

To address this, we have added the following paragraph to Discussion (lines 218-235):

“One of the false negative predictions involved a case of ductal adenocarcinoma. While not excluded from the datasets, this is a cancer subtype our AI model is not specifically trained or validated to detect. Although this is the second most common subtype of prostate cancer after acinar adenocarcinoma, it remains rare, comprising only 0.17% of cases³³. Due to its low prevalence, acquiring sufficient training data for robust AI model development and validation remains a significant challenge. This case also highlights the broader issue of detecting and differentiating various intraductal proliferations such as high-grade prostatic intraepithelial neoplasia (HGPIN), atypical intraductal proliferation (AIP), and intraductal carcinoma (IDC). Our current model is not validated to distinguish these entities, and they fall outside the scope of this study. While IHC—particularly basal-cell markers—can aid in distinguishing IDC from invasive cribriform (Gleason pattern 4) or comedonecrotic (Gleason pattern 5) cancers, current guidelines recommend IHC primarily in cases lacking definitive invasive cancer, which are relatively rare (0.06–0.26% of biopsies)³⁴. Moreover, the diagnostic value of basal-cell IHC in differentiating HGPIN, AIP, and IDC is limited; such assessments continue to rely heavily on expert interpretation of H&E morphology, occasionally supplemented with non–basal-cell markers such as AMACR. As we do not currently report separate performance metrics for ductal adenocarcinoma or intraductal lesions, these limitations further underscore the importance of human oversight and the current role of AI as a decision-support tool rather than a stand-alone diagnostic system. Nevertheless, we aim to expand our dataset to include more such cases in future iterations of the model.”

Comment 2:

Additionally, the study acknowledges that tissue artifacts and stain variation can impact model accuracy. It would be valuable to know whether the authors anticipate improvements in the model’s robustness to such challenges, and whether future iterations will incorporate mechanisms to better handle these limitations.

These are important considerations, and we have addressed them to some extent in our previous publications:

Regarding stain variation, our primary validation demonstrated that the AI system maintained consistent performance across external datasets from multiple laboratories spanning various countries and continents (<https://doi.org/10.48550/arXiv.2502.21264>). Thus, the model seems fairly robust when it comes to variation in staining. There are also color calibration methods designed to standardise the appearance of slides prior to AI model evaluation. A comparative analysis of three such methods—CycleGAN, Macenko, and SIERRA—has been published previously (<https://doi.org/10.1016/j.modpat.2025.100715>).

The issue of tissue artifacts is indirectly addressed in a previous publication, where we propose the use of a conformal prediction framework to quantify the AI model’s confidence in its predictions. This approach allows the AI to flag samples that contain unfamiliar or out-of-distribution regions—such as tissue artifacts—that the model has not been trained to recognize (<https://doi.org/10.1038/s41467-022-34945-8>).

To address this, we have added the following passage to Discussion (lines 245-251):

“While stain variation and tissue artifacts can affect AI performance, our primary validation study²⁹ demonstrated model robustness against stain variation across multiple fully external datasets. Additionally, several color calibration techniques have been proposed to further enhance model robustness to staining variability³⁶. In contrast, tissue artifacts that obscure key morphological features remain a significant challenge. We have previously investigated this issue and plan to implement a conformal prediction framework to flag uncertain predictions—such as those arising from obscured or artifact-laden tissue—in future versions of the model³⁵.”

In conclusion, this AI model represents a compelling advancement in prostate pathology, with the potential to enhance diagnostic efficiency and reduce unnecessary IHC use. It could become a valuable tool for practicing pathologists, supporting more cost-effective and streamlined workflows.

Reviewer #2 (Remarks to the Author):

The work uses an AI model on H&E images of prostate core needle biopsies from three different cohorts to decreased the need for immunohistochemistry.

A well-written, concise, and direct manuscript with impact in the healthcare management.

The work could benefit from minor revisions/clarifications, listed below:

- It is not stated whether the immunohistochemistry technique developed at the three sites is the same (protocol, antibody used – namely the clone, equipment, etc.); the same for the HE staining.

This is an important point, and we agree that our study would benefit from this kind of detailed technical information. Unfortunately, comprehensive data on the immunohistochemistry (IHC) and H&E staining protocols—such as antibody clones, staining equipment, and site-specific procedures—were not available for the contributing centers. This information is particularly difficult to retrieve given that the data were collected several years ago, during which time local protocols, reagents, and equipment may have changed. Nonetheless, given the numerous factors that influence histopathological slide appearance, including differences in laboratory protocols, reagents, equipment, and personnel, it is reasonable to assume a degree of inter-site variability, even if we are unable to quantify this variation.

To clarify this point, we have revised the manuscript text by adding the following sentence at the end of each cohort description in Methods (lines 610-611, 624-625, 646-647):

“Detailed information on the IHC and H&E staining protocols, including antibody clones, equipment, and site-specific procedures, was not available.”

- Details of scanning and navigation are missing, like magnification, and how this data might interfere with both visual and computational analysis.

While details such as scanning magnification and pixel size are not explicitly described in the current manuscript, this information is provided in detail in **Table 2** of our pre-specified study

protocol, which is referenced at the beginning of the “Methods” section (<https://doi.org/10.1101/2024.07.04.24309948>).

We have now added details on scanning magnification and pixel size as part of the current manuscript in Methods (lines 604, 617, 630-631):

“(…) The slides were digitized with a Hamamatsu S60 scanner (40x, pixel size 0.2199 μm).”

“(…) The slides were digitized with a Philips IntelliSite Ultra Fast Scanner (40x, pixel size 0.2500 μm , the same device as for the cohort SCH).”

“(…)The slides were digitized with a Philips IntelliSite Ultra Fast Scanner (40x, pixel size 0.2500 μm , the same device as for the cohort SFR).”

- It would be interesting to try and explain the difference in IHC request reduction across the cohorts based on their origin: with two being close to 40% (SUF and SFR) and one around 20% (SCH). How do you explain this? What factors might have influenced it?

We appreciate the reviewer’s insightful observation regarding the variation in IHC reduction across cohorts. A key contributing factor is the underlying difference in the proportion of benign versus malignant slides: the SUH and SFR cohorts had a higher percentage of benign slides (55.1% and 58.9%, respectively), compared to the SCH cohort (39.6%). Since our approach targets IHC savings specifically on benign cases, cohorts with a larger fraction of benign slides inherently offer greater potential for IHC reduction.

However, even when isolating slides with a benign ground truth, the proportion of IHC savings still differs notably across sites—80.6% for SUH, 71.2% for SFR, and 52.3% for SCH—suggesting that additional factors are at play. We believe these differences likely reflect variations in institutional and individual pathologist IHC ordering practices. For instance, some sites may apply IHC more liberally, even for mildly suspicious morphologies, whereas others might restrict IHC use to cases with stronger atypical features. Such variability in diagnostic thresholds could significantly influence the extent of potential IHC savings.

We have addressed this variation in more detail in the revised manuscript in Discussion (lines 197-207) :

“The observed differences in IHC reduction across cohorts can be partially explained by cohort composition—specifically, the proportion of benign slides. The SUH and SFR cohorts included a higher percentage of benign cases (55.1% and 58.9%, respectively) compared to the SCH cohort (39.6%). Since our approach targets IHC savings exclusively for benign slides, a smaller overall impact in the SCH cohort is expected. However, even when considering only slides with a benign ground truth, the IHC savings vary substantially across sites—80.6% for SUH, 71.2% for SFR, and 52.3% for SCH—indicating that other factors contribute as well. We believe this variation likely reflects differences in institutional and individual IHC ordering practices. For example, some sites may have a lower threshold for initiating IHC, applying it even for mildly suspicious morphologies, while others may reserve IHC for cases with more overt atypia. Such differences in diagnostic thresholds and practice patterns can meaningfully influence the potential for IHC reduction.”

- The review of cases by pathologists is done with the addition of 12 cases for balancing, considering the various ISUP groups – how is this distribution done?

The original review set included 22 false-negative WSIs, i.e. cases where the model suggested omitting IHC, but IHC had in fact been performed during the original diagnostic process. As the reviewing pathologists were aware that all false-negative slides had undergone IHC, there was a risk that this knowledge could influence their assessment. To mitigate this potential bias, we supplemented the review set with 12 additional WSIs, selected randomly from the validation sets, with a balanced ISUP grade distribution: 2 benign cases and 2 cases from each of the ISUP groups 1 through 5. This approach helped obscure the selection criterion and provided a more balanced and representative set of slides for review. These slides were not added for dataset balancing purposes, but rather to obscure the fact that the original cases were exclusively IHC-validated false negatives.

To clarify this point, we have revised the manuscript text as follows in Results (lines 153-154):

“(…) the purpose of adding these slides was not to balance the dataset, but to mask the fact that the original cases were exclusively IHC-validated false negatives. “

- With a threshold of 0.50, 22 false negative WSIs were re-evaluated. How do you interpret the impact of technical quality and control, given that the three high-grade cancer false negative cases presented artifacts?

Thank you for highlighting this important point, which was also noted by another reviewer. At present, the AI model does not include a dedicated module for detecting or flagging technical artifacts in WSIs. However, we acknowledge that artifacts can significantly impact model performance, particularly in high-grade cases, and we consider artifact detection and quality control an important area for future development. Integrating a dedicated artifact detection component is a planned improvement for future iterations of the AI system.

We have previously addressed the issue of tissue artifacts indirectly by proposing a conformal prediction framework designed to assess the model's confidence in its outputs. This approach allows AI models to recognize and flag regions that deviate from the training data distribution, including potential tissue artifacts (<https://doi.org/10.1038/s41467-022-34945-8>).

We have revised the manuscript text as follows to address this in Discussion (lines 248-251):

“(…) tissue artifacts that obscure key morphological features remain a significant challenge. We have previously investigated this issue and plan to implement a conformal prediction framework to flag uncertain predictions—such as those arising from obscured or artifact-laden tissue—in future versions of the model³⁵.”

- It is stated, without additional data, that pathologists confirmed the AI model identified areas of interest – it was not explained how this evaluation was conducted.

Thank you for this important comment. The evaluation of whether the AI model identified diagnostically relevant areas was conducted by having a pathologist assess the full WSIs and annotate the regions considered most suspicious for malignancy in each image. This was done prior to viewing any attention heatmaps from the AI. These expert annotations were then compared both to the AI model's attention maps and to the areas of actual IHC expression. This allowed us

to qualitatively assess the overlap between pathologist-identified regions of interest, the AI's high-attention areas, and the locations where IHC confirmed the presence of cancer or atypia.

We have clarified this methodology in the revised manuscript text in Results (lines 180-186):

“One pathologist (A.B.) independently reviewed all false-negative WSIs in full and annotated the regions deemed most suspicious for malignancy. These initial assessments were then discussed in a consensus meeting with a second pathologist (L.E.), during which the annotated regions were compared to both the corresponding IHC staining patterns (where available) and the AI model's attention maps. This review confirmed a strong correspondence between the pathologist-identified areas of concern, IHC-confirmed regions, and the model's high-attention zones.”

- It seems risky to state that since there are multiple WSIs from the same patient, the probability of false negatives in cancer diagnosis is reduced.

Thank you for your comment, we are happy to clarify this point. Our statement regarding reduced false negative rates at the patient level is based on a probabilistic observation rather than an assumption. Specifically, if the AI model exhibits a given false negative rate (FNR) on a per-slide level, the probability of a patient with N cancer-containing slides being misclassified as negative across all slides is FNR^N , assuming independence across slides. As patients typically have multiple WSIs, the likelihood of all slides being simultaneously misclassified is always equal to or lower than the per-slide FNR.

- Regarding Eloy C's article, it mentions that the 2-week wash-out period is short, but without scientific evidence (the College of American Pathologists uses this period as a reference for WSI validation in a clinical context).

Thank you for pointing this out. Upon further review, we acknowledge that our initial concern regarding the two-week wash-out period being too short was unfounded. As you mention, the College of American Pathologists (CAP) recommends a minimum of two-week wash-out period for whole-slide imaging (WSI) validation studies, based on a systematic review of the literature. We appreciate the reviewer's correction and have revised the manuscript accordingly to reflect this established guideline.

The following has been removed from the manuscript (Discussion, line 299):

~~especially considering the short washout period of only two weeks.~~

More limitations of the work should be mentioned, rather than just the limitations of other studies, such as: low variability in the training set (which will be a difficulty for implementation in other laboratory settings); lack of information regarding laboratorial procedures and eventual variability which can impact the AI model performance; the image processing time, IT infrastructure, integration methods, and maintenance costs for using this AI model were not mentioned.

We fully agree that it is important to acknowledge limitations that may impact real-world implementation. With regard to training data variability, we would like to clarify that our dataset is among the largest used for task-specific prostate AI models to date. In our previous validation study, we demonstrated that training on such a large and well-curated dataset resulted in strong generalization performance across 12 external test sets from 11 different countries (<https://doi.org/10.48550/arXiv.2502.21264>). We believe this highlights the robustness and applicability of our approach across diverse clinical settings.

As for the lack of detail on laboratory procedures and variability, we refer the reader to our pre-specified study protocol (<https://doi.org/10.1101/2024.07.04.24309948>), where this information is described in greater detail.

In terms of image processing time and technical performance, the aforementioned validation paper also provides relevant benchmarks. The average inference time per WSI is on the order of 10-30 seconds depending on the amount of tissue per slide and the hardware the model is deployed on, which is consistent with operational feasibility in clinical workflows.

We also acknowledge the reviewer's valid point regarding the lack of discussion on infrastructure and maintenance costs associated with AI deployment. While this was not within the scope of the present study, we agree that these are critical considerations. We have now added a note in the discussion highlighting the need for future cost-effectiveness analyses that explicitly include IT integration, maintenance, and operating costs. This is an area we hope to explore in future work in collaboration with health economics experts.

We have revised the manuscript text as follows in the Discussion (lines 315-321):

“While our study focuses on diagnostic performance and potential IHC savings, broader implementation of AI in pathology will also require careful consideration of integration costs. This includes infrastructure requirements, image processing times, IT maintenance, and long-term operating expenses. We advocate for future cost-effectiveness studies that comprehensively evaluate these factors alongside diagnostic benefits. Such analyses will be critical in assessing the real-world value of AI-assisted pathology workflows, and we plan to pursue this in collaboration with health economics experts.”

Reviewer #3 (Remarks to the Author):

This is an interesting study that could potentially have significant impact on anatomic pathology laboratory workflow and cost reduction. It provides preliminary findings that are promising. One minor comment I have is that some of the statements that are being made may be on the side of overstatements as the data need further validation (more slides from more institutions, different practice settings, etc.). Additionally, authors should consider discussing whether/how the implementation of such AI in the real-life practice setting will be actually reducing the overall costs (i.e. will the benefit of the AI implementation (reducing # of IHCs) outweigh the cost of AI implementation).

Thank you for your thoughtful feedback and for recognizing the potential impact of this study. We agree that certain claims must be interpreted with appropriate caution, particularly given that the current analysis is based on IHC data from only three cohorts. While these findings are encouraging, further validation across broader datasets and clinical settings is indeed necessary.

That said, we would like to note that our main validation study demonstrated strong generalization performance of the AI model across 12 external cohorts from 11 countries, underscoring the model’s robustness in diverse practice environments. However, we acknowledge that this previous evaluation did not focus specifically on the subset of diagnostically challenging slides requiring IHC. We have revised the manuscript to better reflect these limitations and to temper any overstatements.

The reviewer's point regarding the cost–benefit balance of AI implementation is indeed critical. A similar concern was raised by another reviewer, and we fully agree that a comprehensive cost-effectiveness evaluation is essential to assess the real-world value of integrating AI into pathology workflows. While the present study focuses on diagnostic performance and the potential to reduce IHC usage, it does not encompass the broader economic implications, including costs related to infrastructure, integration, and system maintenance. We have now addressed this limitation in the discussion and emphasized the need for future studies to rigorously evaluate whether the financial benefits of reduced IHC usage outweigh the operational costs of deploying AI. This is a priority for future work, which we plan to pursue in collaboration with health economics experts.

Regarding the need for further validation, we have revised the manuscript text as follows in the Discussion (lines 306-310):

“However, it must be noted that the present analysis is limited to IHC data from three cohorts. While our previous large-scale validation study demonstrated that the AI model generalizes well across 12 external cohorts from 11 countries, those evaluations did not specifically focus on diagnostically challenging cases requiring IHC. This highlights the need for further validation in broader and more diverse clinical settings, particularly for difficult cases.”

Regarding the cost-benefit balance, we have revised the manuscript text as follows in the Discussion (lines 315-321):

“While our study focuses on diagnostic performance and potential IHC savings, broader implementation of AI in pathology will also require careful consideration of integration costs. This includes infrastructure requirements, image processing times, IT maintenance, and long-term operating expenses. We advocate for future cost-effectiveness studies that comprehensively evaluate these factors alongside diagnostic benefits. Such analyses will be critical in assessing the real-world value of AI-assisted pathology workflows, and we plan to pursue this in collaboration with health economics experts.”